# Dopaminergic Modulation of Orofacial Mechanical Hypersensitivity Induced by Infraorbital Nerve Injury

**DOI:** 10.3390/ijms21061945

**Published:** 2020-03-12

**Authors:** Hiroharu Maegawa, Nayuka Usami, Chiho Kudo, Hiroshi Hanamoto, Hitoshi Niwa

**Affiliations:** Department of Dental Anesthesiology, Osaka University Graduate School of Dentistry, Suita, Osaka 565-0871, Japan; n.adachi@dent.osaka-u.ac.jp (N.U.); ckudo@dent.osaka-u.ac.jp (C.K.); hanamoto@dent.osaka-u.ac.jp (H.H.); niwa@dent.osaka-u.ac.jp (H.N.)

**Keywords:** chronic constriction injury, infraorbital nerve, A11 nucleus, dopamine receptor, quinpirole, muscimol, 6-hydroxydopamine, phosphorylated extracellular signal-regulated kinase (pERK), trigeminal spinal subnucleus caudalis

## Abstract

While the descending dopaminergic control system is not fully understood, it is reported that the hypothalamic A11 nucleus is its principle source. To better understand the impact of this system, particularly the A11 nucleus, on neuropathic pain, we created a chronic constriction injury model of the infraorbital nerve (ION-CCI) in rats. ION-CCI rats received intraperitoneal administrations of quinpirole (a dopamine D2 receptor agonist). ION-CCI rats received microinjections of quinpirole, muscimol [a gamma-aminobutyric acid type A (GABA_A_) receptor agonist], or neurotoxin 6-hydroxydopamine (6-OHDA) into the A11 nucleus. A von Frey filament was used as a mechanical stimulus on the maxillary whisker pad skin; behavioral and immunohistochemical responses to the stimulation were assessed. After intraperitoneal administration of quinpirole and microinjection of quinpirole or muscimol, ION-CCI rats showed an increase in head-withdrawal thresholds and a decrease in the number of phosphorylated extracellular signal-regulated kinase (pERK) immunoreactive (pERK-IR) cells in the superficial layers of the trigeminal spinal subnucleus caudalis (Vc). Following 6-OHDA microinjection, ION-CCI rats showed a decrease in head-withdrawal thresholds and an increase in the number of pERK-IR cells in the Vc. Our findings suggest the descending dopaminergic control system is involved in the modulation of trigeminal neuropathic pain.

## 1. Introduction

The modulation of nociceptive inputs from descending noradrenergic and serotonergic control systems is well documented [1,2,3,4]. It is likely that the dopaminergic descending control system is also involved in the processing of nociceptive stimuli [5,6]. Several studies showed that the dopaminergic system was related to antinociception [7,8,9]. However, the descending dopaminergic control system is less well defined [6].

Dopamine receptors are classified into two families: D1-like (D1 and D5) and D2-like (D2, D3, and D4) receptors. The D2-like receptors mediate antinociception [6,10,11] and are the most prominent dopamine receptor subtype in the spinal dorsal horn (SDH) and medullary dorsal horn (MDH) [12,13,14].

The hypothalamic A11 nucleus is the principle source of descending dopaminergic pathways [6]. The A11 nucleus has projections to the MDH [15], and local electrical or pharmacological stimulation of the A11 nucleus inhibits responses of SDH/MDH neurons to noxious stimulation, through the activation of segmental D2 receptors [10,16]. On the other hand, electrical lesions of the A11 nucleus facilitate responses of neurons in the trigeminocervical complex to both noxious and innocuous mechanical stimuli [16]. Administration of quinpirole (a dopamine D2 receptor agonist) to the A11 nucleus attenuates mechanical hypersensitivity in rats with spinal nerve ligation [17]. Stimulation of the A11 nucleus attenuates trigeminal neuropathic pain in mice [18].

However, the function of the A11 nucleus is not fully revealed. The aim of the present study was to assess: (1) the effects of administration of a dopamine D2 receptor agonist on trigeminal neuropathic pain and (2) the effects of the A11 nucleus on trigeminal neuropathic pain to better understand the role of the descending dopaminergic control system. In the present study, we created a chronic constriction injury model of the infraorbital nerve (ION-CCI) in rats as the trigeminal neuropathic pain model. Mechanical stimulation by von Frey filament to the maxillary whisker pad skin was used to generate a behavioral response (the head-withdrawal threshold) and an immunohistochemical response [phosphorylated extracellular signal-regulated kinase (pERK) expression in the superficial layers of the trigeminal spinal subnucleus caudalis (Vc)] following intraperitoneal (i.p.) administration of quinpirole in ION-CCI rats. We used the same methods to evaluate the effects of microinjections of quinpirole, muscimol [a gamma-aminobutyric acid type A (GABA_A_) receptor agonist], or neurotoxin 6-hydroxydopamine (6-OHDA) into the A11 nucleus in ION-CCI rats to assess the role of the A11 nucleus on trigeminal neuropathic pain.

## 2. Results

### 2.1. The Change of Head-Withdrawal Threshold following ION-CCI

Rats received mechanical stimulation by von Frey filaments to the maxillary whisker pad skin on the ipsilateral side to nerve ligation to measure head-withdrawal threshold. The results of the test of the head-withdrawal threshold showed a significant interaction [F (4, 40) = 8.20, *p* < 0.001]. Rats with ION-CCI (*n* = 6) showed mechanical hypersensitivity from 3 to 21 days after nerve ligation; they showed a significant decrease in head-withdrawal thresholds as compared with sham-operated rats on 3, 7, 14, and 21 days after ION-CCI (7 days after: *p* < 0.001; 3 days after: *p* = 0.002; 14, and 21 days after: *p* = 0.001, Figure 1, Appendix A). Rats with ION-CCI showed a significant decrease in head-withdrawal thresholds to mechanical stimulation 3 days after ION-CCI as compared with before ION-CCI; this decrease lasted until 21 days after nerve ligation (3 and 7 days after: *p* < 0.001, 14 and 21 days after: *p* = 0.001, Figure 1). Sham operated rats (*n* = 6) showed no significant change in head-withdrawal threshold (Figure 1).

### 2.2. The Effect of Systemic Administration of Quinpirole on Mechanical Hypersensitivity and pERK Expression in the Superficial Layers of the Vc

The results of the test of the head-withdrawal threshold showed a significant interaction [F (5, 60) = 13.19, *p* < 0.001]. Fourteen days after ION-CCI, ION-CCI rats that received i.p. administration of quinpirole (*n* = 7) showed a significant increase in head-withdrawal thresholds as compared with those before drug administration. The increase was found 20, 60, 120, 180, and 240 min after the administration of quinpirole (20, 60, 120, and 180 min after: *p* < 0.001; 240 min after: *p* = 0.001; Figure 2A; Appendix A). On the other hand, ION-CCI rats that received i.p. administration of saline (*n* = 7) showed no significant change of head-withdrawal threshold as compared with those before saline administration. The head-withdrawal threshold of ION-CCI rats with i.p. administration of quinpirole was larger than that of ION-CCI rats with i.p. administration of saline at 20, 60, 120, 180, and 240 min after administration, respectively (20, 60, 120, and180 min after: *p* < 0.001; 240 min after: *p* = 0.001; Figure 2A). Twenty minutes after quinpirole or saline administration, we also applied mechanical stimulation by von Frey filament (15 g, 1 Hz, 5 min) to the maxillary whisker pad skin on the ipsilateral side to nerve ligation in ION-CCI rats (*n* = 7 in each group). Five minutes after the stimulation, rats were perfused, and then, immunohistochemical staining for pERK in the Vc was performed (Figure 2B, Appendix A). The greatest number of labeled cells that were pERK immunoreactive (pERK-IR) were found in the superficial layers of the Vc 1.4 mm caudal to the obex in both groups. The number of pERK-IR cells/section in the superficial layers of the Vc of ION-CCI rats that received quinpirole administrations was smaller than that of ION-CCI rats that received saline administrations (16.8 ± 2.4 vs. 27.7 ± 2.7, df = 68, *t* = 7.36, *p* < 0.001, Figure 2C, Appendix A).

### 2.3. The Effect of Microinjections of Quinpirole and Muscimol into the A11 Nucleus on Mechanical Hypersensitivity and pERK Expression in the Superficial Layers of the Vc

Seven days after ION-CCI, a metal cannula was implanted into the skull of ION-CCI rats for a microinjection of drugs. ION-CCI rats that had a metal cannula implanted into the skull exhibited no significant change in head-withdrawal threshold between before (7 days after ION-CCI) and after the implantation procedure (13 days after ION-CCI) (Appendix A).

Fourteen days after ION-CCI (7 days after the implantation), we administered quinpirole or muscimol or saline through the cannula (Figure 3A, Appendix A). Injection sites were distributed −3.3 mm to −3.9 mm from bregma (Figure 3B). The number of misplacements of cannula were as follows: ION-CCI rats with saline microinjection: 2, ION-CCI rats with quinpirole microinjection: 2, and ION-CCI rats with muscimol microinjection: 3. The results of the test of the head-withdrawal threshold showed a significant interaction [F (6, 54) = 14.38, *p* < 0.001]. The head-withdrawal threshold in ION-CCI rats that received microinjections of quinpirole (*n* = 7) was larger than that in ION-CCI rats that received microinjections of saline (*n* = 7) 20 and 40 min after the microinjection (20 min: *p* < 0.001, 40 min: *p* = 0.004, Figure 3C). The head-withdrawal threshold in ION-CCI rats that received microinjections of muscimol (*n* = 7) was larger than that in ION-CCI rats that received microinjections of saline 20 and 40 min after the microinjection (*p* < 0.001, Figure 3C). ION-CCI rats that received microinjections of quinpirole showed a significant increase in head-withdrawal thresholds 20 and 40 min after microinjections as compared with thresholds before microinjections (*p* < 0.001, Figure 3C, Appendix A). ION-CCI rats that received microinjections of muscimol also showed a significant increase in head-withdrawal thresholds 20 and 40 min after microinjections as compared with thresholds before microinjections (*p* < 0.001, Figure 3C). Twenty minutes after quinpirole, muscimol, or saline microinjection, we also applied mechanical stimuli by von Frey filament (15 g, 1 Hz, 5 min) to ION-CCI rats as described above (n = 7 in each group). As with the i.p. administrations, after mechanical stimulation and immunohistochemistry, the greatest number of pERK-IR cells were found in the superficial layers of the Vc 1.4 mm caudal to the obex in all groups (Figure 3D, Appendix A). As a result of a test of the number of pERK-IR cells in the Vc, the main effect was found to be significant [F (2, 102) = 64.17, *p* = 0.001]. The mean number of pERK-IR cells/section in the Vc of ION-CCI rats that received microinjections of quinpirole was smaller than that of ION-CCI rats that received microinjections of saline (15.9 ± 0.4 vs. 27.0 ± 0.7, *p* < 0.001, Figure 3E, Appendix A). The mean number of pERK-IR cells/section in the Vc of ION-CCI rats that received microinjections of muscimol was also smaller than that of ION-CCI rats that received microinjections of saline (12.9 ± 1.7 vs. 20.7 ± 0.7, *p* < 0.001, Figure 3E).

### 2.4. The Effect of Microinjections of 6-OHDA into the A11 Nucleus on Mechanical Hypersensitivity and pERK Expression in the Superficial Layers of the Vc

Seven days after ION-CCI, ION-CCI rats received microinjections of 6-OHDA or saline into the A11 nucleus. Fourteen days after 6-OHDA microinjection (21 days after ION-CCI), immunohistochemical staining for tyrosine hydroxylase (TH) was performed in all the rats that were microinjected with 6-OHDA or saline into the A11 nucleus (*n* = 14 in each group, Figure 4A). Injection sites were distributed -3.3 mm to -3.9 mm from bregma (Figure 4B). The number of misplacements of cannula were as follows: ION-CCI rats with saline microinjection: 3 and ION-CCI rats with 6-OHDA microinjection: 4. ION-CCI rats that received 6-OHDA or saline microinjections showed TH immunoreactive (TH-IR) cells throughout the entire rostrocaudal extent of the A11 nucleus. The number of TH-IR cells was large in the rostral part of the A11 nucleus and gradually decreased toward the caudal end. As a result of a test of the number of TH-IR cells in the A11 nucleus, a significant interaction was found [F (1, 52) = 18.13, *p* < 0.001]. In ION-CCI rats with 6-OHDA microinjections, the significant decrease of TH-IR cells in the A11 nucleus was found in the ipsilateral side to 6-OHDA microinjection as compared with the contralateral side (11.9 ± 1.2 vs. 24.3 ± 1.0, *p* < 0.001, Figure 4C, Appendix A). ION-CCI rats with 6-OHDA microinjections showed approximately 40% fewer TH-IR cells in the A11 nucleus on the ipsilateral side to microinjections as compared with the ipsilateral side in ION-CCI rats with saline microinjections (11.9 ± 1.2 vs. 29.2 ± 1.0, *p* < 0.001, Figure 4C). The results of the test of the head-withdrawal threshold showed a significant interaction [F (1, 12) = 10.8, *p* < 0.01]. At 14 days after 6-OHDA microinjection (21 days after ION-CCI), ION-CCI rats that received microinjections of 6-OHDA into the A11 nucleus (*n* = 7) showed a significant decrease in head-withdrawal thresholds as compared with thresholds of 7 days after ION-CCI (before 6-OHDA microinjection) (*p* < 0.001, Figure 4D, Appendix A). At 21 days after ION-CCI, the head-withdrawal threshold of ION-CCI rats that received microinjections of 6-OHDA into the A11 nucleus was smaller than that of ION-CCI rats that received microinjections of saline (*n* = 7, *p* = 0.004, Figure 4D). We also applied mechanical stimuli by von Frey filament (15 g, 1 Hz, 5 min) to ION-CCI rats as described above (*n* = 7 in each group). The greatest number of pERK-IR cells were found in the superficial layers of the Vc 1.4 mm caudal to the obex in both groups (Figure 4E, Appendix A). The mean number of pERK-IR cells/section in the Vc was larger in ION-CCI rats that received microinjections of 6-OHDA as compared with ION-CCI rats that received microinjections of saline (32.4 ± 6.6 vs. 26.0 ± 2.1, df = 68, *t* = 3.06, *p* < 0.05, Figure 4F, Appendix A).

## 3. Discussion

We found that the systemic administration of a dopamine D2 receptor agonist attenuated the mechanical hypersensitivity induced by ION-CCI and decreased the number of pERK-IR cells in the superficial layers of the Vc in rats with ION-CCI. Following stimulation, pERK expression reaches to a peak within 3 min [19]. Previously, ION-CCI rats were perfused 5 min after stimulation, and pERK expression in the Vc was investigated [20]. Therefore, we perfused rats 5 min after mechanical stimulation to the maxillary whisker pad skin according to the previous report [20]. Our findings indicate that systemic administration of a dopamine D2 receptor agonist suppresses mechanical hypersensitivity induced by ION-CCI. Previously, systemic administration of dopamine D2 or D2/D3 receptor agonists was reported to attenuate pain-related behavior in control [21,22,23,24,25] and hypersensitivity conditions induced by intraplanter injection of carrageenan [26] and sciatic nerve ligation [27]. Previous work has shown that neuropathic pain rat models that received i.p. administration of quinpirole showed an increase in withdrawal threshold to mechanical stimuli [27]. While the applied drug dose was the same in both studies, the methodology was different with the previous study using rats with sciatic nerve ligation, as opposed to our rats that had infraorbital nerve ligation; in addition, different measurement techniques were used. This might explain why the previous work reported an increase in withdrawal threshold that persisted for more than 24 h, but our rats with ION-CCI showed an increase in head-withdrawal threshold for 180 min.

The A11 nucleus is known to have a high density of dopamine D2 receptors [28]. In the present study, we found that microinjections of a dopamine D2 receptor agonist into the A11 nucleus attenuated the mechanical hypersensitivity induced by ION-CCI and decreased the number of pERK-IR cells in the Vc in ION-CCI rats. Our results show that microinjections of a dopamine D2 receptor agonist into the A11 nucleus change the activity of the A11 nucleus and inhibit the activity of the Vc neurons in ION-CCI rats, suggesting that the A11 nucleus is involved in descending modulation of mechanical hypersensitivity induced by ION-CCI through dopamine D2 receptors. Microinjection of a dopamine D2 receptor agonist into the A11 nucleus attenuated mechanical hypersensitivity induced by spinal nerve ligation in a previous study [17]; in our work, we found that the duration of the increase in head-withdrawal threshold to mechanical stimuli was longer in ION-CCI rats that received i.p. administration of quinpirole as compared with ION-CCI rats that received microinjections of quinpirole directly into the A11 nucleus. The presence of bilateral projections from the A11 nucleus to MDH is reported [15]. The expression of dopamine D2 receptors is indicated in the Vc [13,18,29]. Additionally, microinjections of quinpirole into the MDH is reported to elicit the depression of c-fiber evoked responses [29]; this finding suggests the inhibition of Vc neurons by the activation of dopamine D2 receptors in the Vc. Taken together, these findings suggest that dopamine released from dopaminergic neurons in the A11 nucleus modulate the activity of the Vc neurons through dopamine D2 receptors expressed in the Vc.

Administration of dopamine D2 receptor agonists to regions external to the Vc has been shown to suppress neuropathic pain. Striatal administration of dopamine D2 receptor agonist was reported to attenuate withdrawal responses to mechanical stimuli in rats with peroneal and tibial nerve ligations [30]. Intrathecal administration of a dopamine D2 receptor agonist was reported to inhibit allodynic responses to mechanical stimuli in rats with sciatic nerve ligation [27]. Thus, it is likely that when dopamine D2 receptor agonists are administered intraperitoneally, they can act thorough not only the A11 nucleus but also other regions, such as the striatum or intrathecal space. Alleviation of mechanical hypersensitivity following i.p. administration of dopamine D2 receptor agonists may therefore be a mixture of effects that occur through these regions.

Our findings suggest that microinjections of a GABA_A_ receptor agonist into the A11 nucleus change the activity of the A11 nucleus and inhibit the activity of Vc neurons in ION-CCI rats. These findings also suggest that the A11 nucleus is involved in descending modulation of trigeminal neuropathic pain. Microinjections of a GABA_A_ receptor agonist into the A11 nucleus have been previously shown to attenuate mechanical hypersensitivity induced by spinal nerve ligation [17]; microinjections of a GABA_A_ receptor agonist into the A11 nucleus also reduced rubbing responses to subcutaneous formalin injection into the upper lip of rats [15]. We found that ION-CCI rats that received microinjections of 6-OHDA into the A11 nucleus exhibited a decrease in head-withdrawal thresholds and an increase in the number of pERK-IR cells in the Vc. These findings suggest that the destruction of dopaminergic cells in the A11 nucleus exacerbates mechanical hypersensitivity induced by ION-CCI. Inhibition of the A11 nucleus by muscimol alleviated mechanical hypersensitivity induced by ION-CCI and the destruction of dopaminergic cells by 6-OHDA in the A11 nucleus exacerbated mechanical hypersensitivity. In the A11 nucleus, there are three types of neurons: dopamine-, GABA-, and calcitonin gene-related peptide (CGRP)-containing neurons [31,32]. Taken together, these results suggest that GABA- or CGRP-containing neurons in the A11 nucleus may be involved in the alleviation of mechanical hypersensitivity induced by ION-CCI. The A11 nucleus contains almost 2.5 times more GABA-containing neurons as compared with dopamine-containing neurons, and noxious facial stimulation activates GABA-containing neurons in the A11 nucleus [15], suggesting that GABA-containing neurons in the A11 nucleus might be involved in the regulation of dopaminergic neuron activity. Dopamine D2 agonist and GABA_A_ receptor agonist were reported to affect motor function [33,34,35,36]. Therefore, the change of head-withdrawal threshold measured in our study might be the result of not only attenuation of mechanical hypersensitivity but also change of motor function by quinpirole and muscimol. This point could be a limitation of the present study.

Microinjections of 6-OHDA into the A11 nucleus were reported to decrease face rubbings induced by formalin injection into the upper lip of normal rats [15] and increase head-withdrawal threshold to mechanical stimuli with von Frey filament in ION-CCI mice [18]. The former showed approximately 40% decrease in TH-IR cells, and the latter showed approximately 70% decrease. Dopamine is known to have a higher affinity to D2-like receptors and activates them with lower concentrations of dopamine than D1-like receptors [37]. Therefore, low levels of dopamine could activate D2-like receptors leading to inhibition of pain, and high levels of dopamine would activate D1-like receptors leading to exacerbation of pain [38]. Furthermore, even high levels of dopamine could activate D2-like receptors due to a shift of preference to D2-like receptors [37,39]. Protein of dopamine D2 receptors are increased in the dorsal horn of rats with spinal nerve ligation [40], which suggests that a change in expression of dopamine receptors may occur between normal and neuropathic pain model rats. For our study, as well as others [15,18], the number of TH-IR neurons, not dopamine levels, was investigated, so it remains unclear whether the decrease in TH-IR neurons reflects the decrease in dopamine levels.

A previous report indicated that mechanical hypersensitivity induced by ION-CCI lasted 42 days after nerve ligation [41]. Heat hypersensitivity was also reported [27,42], and it lasted 12 days after ION-CCI [42]. A previous report also indicated that intrathecal coadministration of quinpirole and µ-opioid receptor agonist attenuated mechanical and heat hypersensitivity induced by sciatic nerve ligation [43]. Increases of dopamine D2 receptors and their colocalization with µ-opioid receptors were reported in the dorsal horn during pathological process of pain [44,45]. A similar reaction may occur in the Vc of ION-CCI rats.

In summary, we showed that dopamine D2 agonist attenuated mechanical hypersensitivity induced by ION-CCI. We suggested that the A11 nucleus is involved in the modulation of mechanical hypersensitivity induced by infraorbital nerve ligation in rats through a modulation of activities of the Vc neurons. These findings help uncover the characteristics of the descending dopaminergic system and may lead to further alleviation of neuropathic pain.

## 4. Material and Methods

Protocols in the present study were performed in accordance with the ethical guidelines of the International Association for the Study of Pain [46] and were approved by the Osaka University Graduate School of Dentistry Animal Care and Use Committee (29-003, 9 May 2017).

### 4.1. Infraorbital Nerve Constriction

Male Wistar rats (SLC, Hamamatsu, Japan) were housed under a 12 h dark/light cycle with ad libitum access to food and water. Rats ranged from 170 to 200 g at the beginning of the experiment and were anesthetized by i.p. administration of saline solution mixed with 2.0 mg/kg midazolam (Sandoz, Tokyo, Japan), 2.5 mg/kg butorphanol (Meiji Seika Pharma, Tokyo, Japan), and 0.375 mg/kg medetomidine (Zenoaq, Fukushima, Japan). Rats were placed in the supine position and a small intra oral incision was made along the gingiva-buccal margin proximal to the first molar, exposing the left ION [42]. The ION was ligated loosely with two ligatures using 4-0 silk to create a chronic constriction injury of the ION. For control animals, a sham operation was performed without nerve ligation.

### 4.2. Measurement of Mechanical Head-Withdrawal Threshold

Rats were trained every day to keep their snout protruding from a plastic cage with a small hole on the front wall during mechanical stimulation to the maxillary whisker pad skin. Rats could escape from applied stimulation freely. Once training has been successfully completed, mechanical head-withdrawal threshold was measured using von Frey filaments. Applied mechanical stimuli with von Frey filaments were 0.008, 0.02, 0.04, 0.07, 0.16, 0.4, 0.6, 1, 1.4, 2, 4, 6, 8, 10, 15, 26, 35, 45, and 60 g (North Coast Medical, Morgan Hill, CA, USA). The maximum force of 60 g was used to prevent tissue damage [47]. Mechanical stimuli were applied to the maxillary whisker pad skin of rats on the ipsilateral side to ION-CCI. Head-withdrawal threshold was determined by the minimum force that elicited a withdrawal response to greater than three out of five stimuli. We used six rats with ION-CCI to investigate the time course of mechanical hypersensitivity following ION-CCI. Measurements of the head-withdrawal threshold were determined before and 3, 7, 10, 14, and 21 days after nerve ligation. Rats with ION-CCI and a head-withdrawal threshold below 6 g at 7 days after nerve ligation were categorized as ION-CCI rats. For the subsequent experiments, performed 7 to 21 days after nerve ligation, we used rats that showed a decrease in head-withdrawal threshold (below 6 g) at 7 days after ION-CCI. Seven days after ION-CCI, 42 rats were implanted with metal cannulas to the skull for microinjection of drugs, and 28 rats received microinjections of 6-OHDA into the A11 nucleus following the measurement of the head-withdrawal threshold. For rats that received the implantation of a cannula, we measured head-withdrawal threshold 13 days after ION-CCI, just before subsequent experiments were performed 14 days after ION-CCI. These rats still showed a decrease in head-withdrawal threshold (below 6 g) 13 days after ION-CCI. Measurement of head-withdrawal threshold was performed using a blind protocol, so that the investigator measuring head-withdrawal thresholds was unaware of which experimental procedures had been performed.

### 4.3. Administration of Quinpirole and Muscimol

To assess the effects of administration of a dopamine D2 receptor agonist on the mechanical hypersensitivity induced by ION-CCI, an i.p. administration of 1 mg/kg quinpirole (a dopamine D2 receptor agonist, Sigma, St. Louis, MO, USA) or saline was performed 14 days after nerve ligation [27]. The head-withdrawal thresholds were measured before and 20, 60, 120, 180, and 240 min after the drug administration.

To assess the effects of the change of A11 nucleus activity on the mechanical hypersensitivity induced by ION-CCI, quinpirole or muscimol (a GABA_A_ receptor agonist, Sigma, St. Louis, MO, USA) was microinjected into the A11 nucleus 14 days after nerve ligation. At 7 days after ION-CCI, ION-CCI rats were anesthetized by i.p. administration of saline solution mixed with 2.0 mg/kg midazolam, 2.5 mg/kg butorphanol, and 0.375 mg/kg medetomidine and then placed in a stereotaxic apparatus (Narishige, Tokyo, Japan) with the incisor bar set 3.3 mm below the level of the ear bars. A tiny hole was made in the skull with a drill, and a metal 26-gauge stainless steel guide cannula (Bio Research Center, Aichi, Japan) was implanted for microinjection of drugs into the A11 nucleus. Stereotaxic coordinates were 3.9 mm caudal to the bregma, 0.5 mm left of the midline, and 6 mm ventral to the dural surface, which was 1 mm above the A11 injection site [15]. Screws were placed around the cannula into the skull and the cannula was affixed with dental cement. A dummy cannula was inserted into the guide cannula to ensure patency until the time of microinjection. Seven days after the implantation of the cannula, quinpirole (20 µg in 0.25 µL), muscimol (0.07µg in 0.25 µL), or saline (0.25 µL) was microinjected into the left A11 nucleus [15]. The dummy cannula was removed, and a 33-gauge internal cannula was inserted into the guide cannula. Solutions were administered through the internal cannula connected to a 1-µL Hamilton syringe via a polyethylene tube. The tip of the internal cannula was placed 3.9 mm caudal to the bregma, 0.5 mm left of the midline, and 7 mm ventral to the dural surface—it protruded 1 mm from the end of the guide cannula. The microinjections were performed slowly over a 1-min period. Head-withdrawal thresholds were measured before and 20, 40, and 60 min after drug administration. Measurement of head-withdrawal thresholds was performed using a blind protocol, so that the investigator measuring head-withdrawal thresholds was unaware of which experimental procedures had been performed.

### 4.4. Administration of 6-OHDA and Measurement of Head-Withdrawal Threshold

To assess the effect of destruction of dopaminergic neurons in the A11 nucleus on mechanical hypersensitivity induced by ION-CCI, 6-OHDA was microinjected into the A11 nucleus. Seven days after nerve ligation, 6-OHDA was injected into the A11 nucleus of ION-CCI rats. Thirty minutes prior to surgery, ION-CCI rats received an i.p. injection of desipramine hydrochloride (20 mg/kg, Sigma, St. Louis, MO, USA) to protect noradrenergic neurons and fibers [48,49]. ION-CCI rats then were anesthetized by i.p. administration of saline solution mixed with 2.0 mg/kg midazolam, 2.5 mg/kg butorphanol, and 0.375 mg/kg medetomidine and placed in a stereotaxic apparatus (Narishige, Tokyo, Japan) with the incisor bar set 3.3 mm below the level of the ear bars. Next, 2 mg of 6-OHDA (Sigma, St. Louis, MO, USA) in 1 mL of sterile saline containing 0.01% ascorbic acid was injected into the left A11 nucleus [15,48,49]. Stereotaxic coordinates for the 6-OHDA injection site were 3.9 mm caudal to the bregma, 0.5 mm left of the midline, and 7 mm ventral to the dural surface. The 6-OHDA solution was administered through a 30-gauge cannula with a microinjection pump (Nihon Kohden, Tokyo, Japan) at 1 µL/min for 1 min; the cannula was left in place for 5 min after completion of the injection. The same amount of saline was injected into ION-CCI rats for control.

Fourteen days after the microinjection of 6-OHDA or saline into the A11 nucleus of ION-CCI rats, head-withdrawal thresholds were measured using a blind protocol, so that the investigator measuring head-withdrawal thresholds was unaware of which experimental procedures had been performed.

### 4.5. Immunohistochemical Analysis

To investigate the activity of the Vc neurons, immunohistochemical staining for pERK was performed. Twenty minutes after the i.p. administration of quinpirole or saline as described above, ION-CCI rats received mechanical stimulation with von Frey filaments (15 g, 1 Hz, 5 min) under deep anesthesia with pentobarbital (80 mg/kg). Mechanical stimuli were applied to the maxillary whisker pad skin on the ipsilateral side to ION-CCI. Twenty minutes after microinjection of quinpirole, muscimol, or saline into the A11 nucleus as described above, ION-CCI rats also received similar mechanical stimuli under pentobarbital anesthesia (*n* = 7 in each group), as did ION-CCI rats that were given microinjections of 6-OHDA (*n* = 7 in each group). Five minutes after mechanical stimuli, ION-CCI rats were perfused transcardially with 100 mL of 0.02 M phosphate-buffered saline (PBS, pH 7.4) followed by 300 mL of 4% paraformaldehyde in a 0.1 M phosphate buffer (PB, pH 7.4). Brains were removed and post-fixed overnight in the same solution and then immersed in 30% sucrose in 0.1 M PB. Serial transverse sections of the brainstem were made using a freezing microtome at a thickness of 50 µm. Sections were incubated in 0.3% hydrogen peroxide in methanol for 20 min, immersed in 1% normal goat serum (Vector Labs, Burlingame, CA, USA) for 30 min, and incubated with antipERK antibody (Sigma, St. Louis, MO, USA; diluted 1:1000) for 12 h. Sections were incubated with biotinylated rabbit antigoat antibody (Vector Labs; diluted 1:200) for 1 h and with avidin–biotin–peroxidase complex (Vectastain ABC Elite Kit, Vector Labs, Burlingame, CA, USA) for 1 h; sections then were reacted with 0.05% diaminobenzidine tetrahydrochloride, 0.1% ammonium nickel sulphate, and 0.01% hydrogen peroxide in 0.05 M Tris-HCl buffer (pH 7.2). Sections were mounted on gelatin-coated glass slides, air-dried, and cover-slipped. As immunonegative controls, sections were processed as described above but without applying primary antibody. We examined pERK-IR cells in the superficial layers of the Vc under light microscopy. For every third section, the number of pERK-IR cells was counted in sections from the obex to a section 5 mm caudal to it. The five consecutive sections that contained the greatest number of pERK-IR cells in the Vc were selected in each rat (35 sections in each group), and the mean number of pERK-IR cells/section was calculated.

To detect dopaminergic degeneration in the A11 nucleus following 6-OHDA microinjection, immunohistochemical staining for TH was performed in ION-CCI rats with 6-OHDA microinjection. Serial coronal sections containing the A11 cell group were made using a freezing microtome at a thickness of 50 µm. Sections were incubated in 0.3% hydrogen peroxide in methanol for 20 min, immersed in 1% normal horse serum (Vector Labs, Burlingame, CA, USA) for 30 min, and then incubated with anti-TH antibody (Sigma, St. Louis, MO, USA; diluted 1:8000) for 12 h. Sections were then incubated with biotinylated horse antimouse antibody (Vector Labs, Burlingame, CA, USA; diluted 1:200) for 1 h and with avidin–biotin–peroxidase complex (Vectastain ABC Elite Kit, Vector Labs) for 1 h; sections then were reacted with 0.05% diaminobenzidine tetrahydrochloride, 0.1% ammonium nickel sulphate, and 0.01% hydrogen peroxide in 0.05 M Tris-HCl buffer (pH 7.2). Sections were mounted on gelatin-coated glass slides, air-dried, and cover-slipped. As immunonegative controls, the sections were processed with the same reagents but without applying primary antibody. For every third section, the number of TH-IR cells in the A11 nucleus were counted and the mean number of TH-IR cells/section was calculated. A blind protocol was used, so that the investigator who counted pERK-IR and TH-IR cells was unaware of the experimental procedures performed.

### 4.6. Statistical Analysis

Behavioral data are shown as box and whisker plots. The bottom and top of the box indicate the first and third quartiles, respectively. The end of the whiskers represents the minimum and maximum values of all data. If no median is shown in the box, the median is at the top or bottom of the box. If whisker is not displayed in the box, the maximum and minimum values match the first and third quartiles. Other data are shown as means ± SEM. SPSS statistics (ver. 24, IBM, Chicago, IL, USA) was used for statistical analyses. Differences of head-withdrawal thresholds were analyzed using a two-way repeated measures analysis of variance (ANOVA) followed by Bonferroni corrections for multiple comparisons as post hoc comparison. For the evaluation of immunohistochemical data, we performed a two-way ANOVA followed by Bonferroni tests for multiple comparisons (TH-IR cells), a one-way ANOVA followed by Bonferroni tests for multiple comparisons (pERK-IR cells in the rats with microinjection of quinpirole or muscimol into the A11 nucleus), and a t-test (pERK-IR cells in rats with i.p. drug administration and microinjection of 6-OHDA into the A11 nucleus). *p* < 0.05 was considered statistically significant.

## Figures and Tables

**Figure 1 ijms-21-01945-f001:**
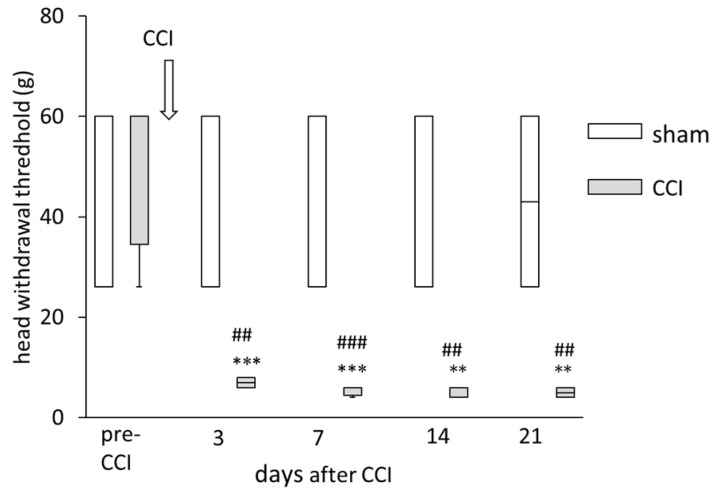
The time course of changes in head-withdrawal threshold of chronic constriction injury of the infraorbital nerve (ION-CCI) in rats. Data are shown as box and whisker plots. The bottom and top of the box indicate the first and third quartiles, respectively. The end of the whiskers represents the minimum and maximum values of all data. Rats with ION-CCI showed mechanical hypersensitivity from 3 to 21 days after nerve ligation; they showed a significant decrease in head-withdrawal thresholds as compared with sham-operated rats on 3, 7, 14, and 21 days after ION-CCI. Rats with ION-CCI showed a significant decrease in head-withdrawal thresholds to mechanical stimulation 3 days after ION-CCI as compared with before ION-CCI; this decrease lasted until 21 days after nerve ligation. *** *p* < 0.001 compared with head-withdrawal threshold before ION-CCI. ** *p* < 0.01 compared with head-withdrawal threshold before ION-CCI. ^###^
*p* < 0.001 compared with head-withdrawal threshold of sham-operated rats on the same time. ^##^
*p* < 0.01 compared with head-withdrawal threshold of sham-operated rats on the same time.

**Figure 2 ijms-21-01945-f002:**
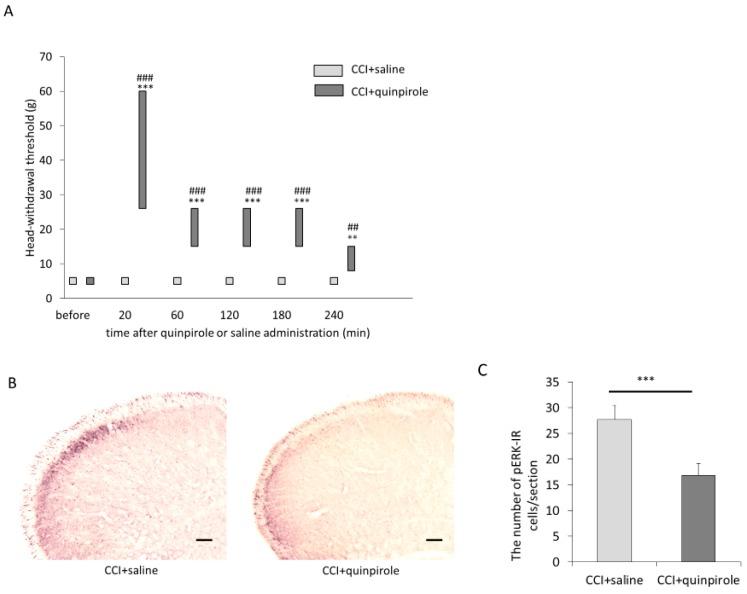
Effects of intraperitoneal (i.p.) administration of quinpirole (1 mg/kg) on ION-CCI. (**A**) The change in head-withdrawal threshold after quinpirole administration. Data are shown as box and whisker plots. The bottom and top of the box indicate the first and third quartiles, respectively. The end of the whiskers represents the minimum and maximum values of all data. Fourteen days after ION-CCI, ION-CCI rats that received i.p. administration of quinpirole showed a significant increase in head-withdrawal thresholds 20, 60, 120, 180, and 240 min after the administration of quinpirole as compared with thresholds before drug administration. *** *p* < 0.001 compared with head-withdrawal threshold before administration. ** *p* < 0.01 compared with head-withdrawal threshold before administration. The head-withdrawal threshold of ION-CCI rats with i.p. administration of quinpirole was larger than that of ION-CCI rats with i.p. administration of saline at 20, 60, 120, 180, and 240 min after administration. ^###^
*p* < 0.001 compared with head-withdrawal threshold of ION-CCI rats that received saline administration at the same time. ^##^
*p* < 0.01 compared with head-withdrawal threshold of ION-CCI rats that received saline administration at the same time. (**B**) Photomicrograph of phosphorylated extracellular signal-regulated kinase (pERK) in the superficial layers of the trigeminal spinal subnucleus caudalis (Vc) of ION-CCI rats that received saline or quinpirole administrations. Scale bar represents 100 µm. (**C**) The number of pERK immunoreactive (pERK-IR) cells/section in the Vc (mean ± SEM). The number of pERK-IR cells/section in the Vc of ION-CCI rats that received quinpirole administrations was significantly larger than that of ION-CCI rats that received saline administrations. *** *p* < 0.001.

**Figure 3 ijms-21-01945-f003:**
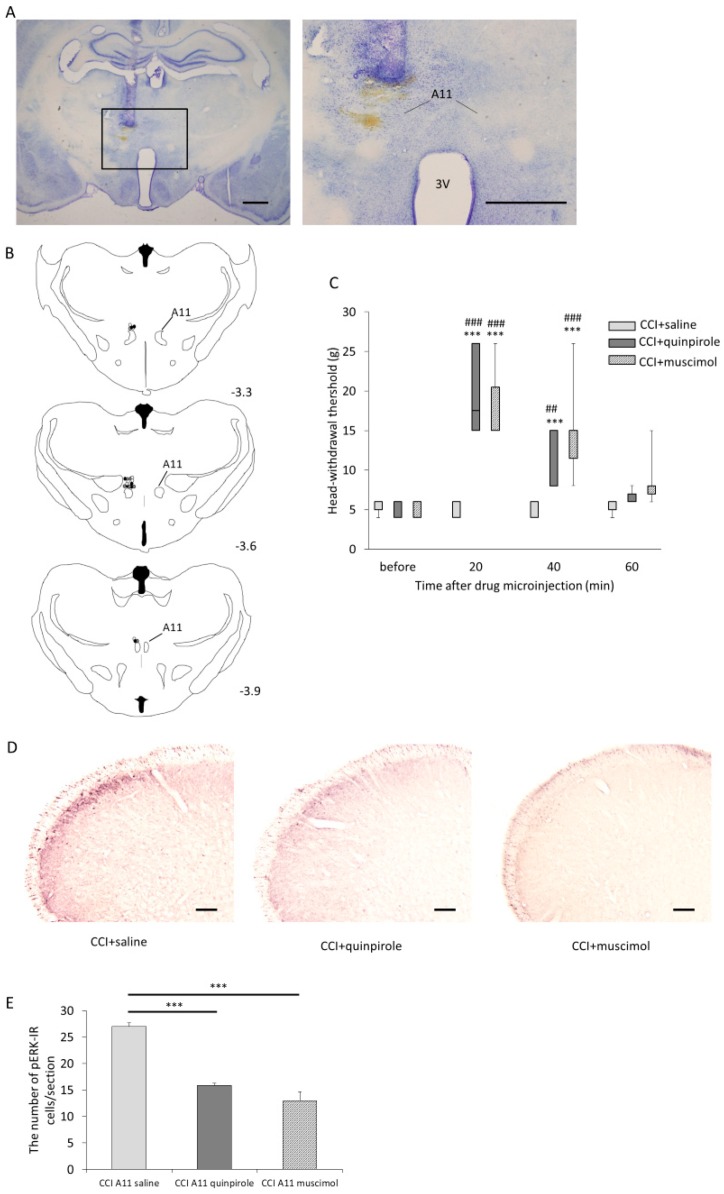
Effects of microinjection of quinpirole (20 µg in 0.25 µL) and muscimol (0.07 µg in 0.25 µL) into the A11 nucleus of ION-CCI. (**A**) Photomicrograph of the injection site within the A11 nucleus at low and high magnification. Scale bar represents 1 mm. 3V: third ventricle. (**B**) Injection site into the A11 nucleus (white circle: saline, gray circle: quinpirole, black circle: muscimol). Digit represents a distance from bregma (mm). (**C**) The change in head-withdrawal threshold after microinjections of quinpirole or muscimol. Data are shown as box and whisker plots. The bottom and top of the box indicate the first and third quartiles, respectively. The end of the whiskers represents the minimum and maximum values of all data. ION-CCI rats that received microinjections of quinpirole showed a significant increase in head-withdrawal threshold 20 and 40 min after the microinjection, as compared with thresholds before the microinjection. ION-CCI rats that received microinjections of muscimol showed a significant increase in head-withdrawal threshold 20 and 40 min after the microinjection as compared with thresholds before microinjections. The head-withdrawal threshold in ION-CCI rats that received microinjections of quinpirole was larger than that in ION-CCI rats that received microinjections of saline 20 and 40 min after the microinjection. The head-withdrawal threshold in ION-CCI rats that received microinjections of muscimol was larger than thresholds in ION-CCI rats that received microinjections of saline 20 and 40 min after the microinjection. *** *p* < 0.001 compared with head-withdrawal threshold before microinjections. ** *p* < 0.01 compared with head-withdrawal threshold before microinjections. ^###^
*p* < 0.001 compared with head-withdrawal threshold of ION-CCI rats that received saline microinjections at the same time. ^##^
*p* < 0.01 compared with head-withdrawal threshold of ION-CCI rats that received saline microinjections at the same time. (**D**) Photomicrographs of pERK-IR cells in the Vc of ION-CCI rats that received microinjections of saline, quinpirole, or muscimol into the A11 nucleus. Scale bar represents 100 µm. (**E**) The number of pERK-IR cells/section in the Vc (mean ± SEM). The number of pERK-IR cells/section in the Vc of ION-CCI rats that received microinjections of quinpirole was smaller than that of ION-CCI rats that received microinjections of saline. The number of pERK-IR cells/section in the Vc of ION-CCI rats that received microinjections of muscimol was smaller than that of ION-CCI rats that received microinjections of saline. *** *p* < 0.001.

**Figure 4 ijms-21-01945-f004:**
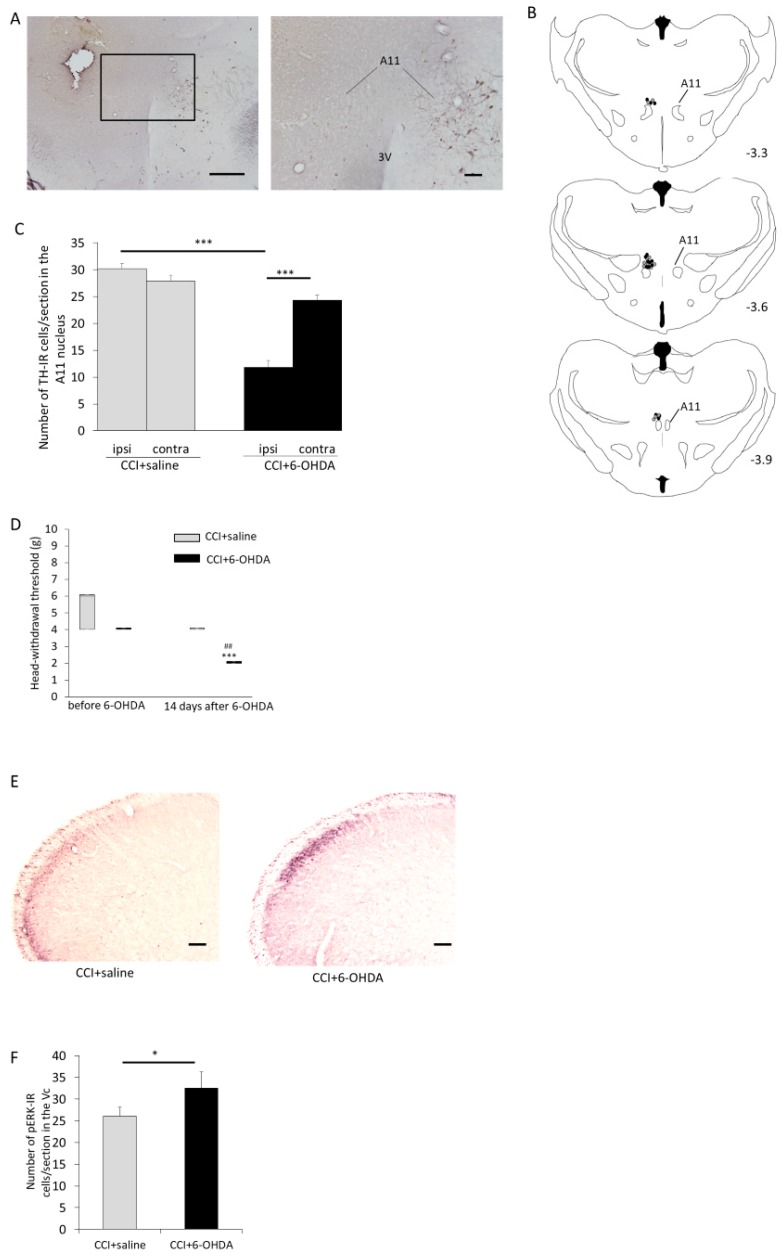
Effects of microinjection of 6-OHDA (2 mg/mL, 1 µL/min for 1 min) into the A11 nucleus on ION-CCI. (**A**) Photomicrograph of the A11 nucleus at low and high magnification. The immunoreactivity of tyrosine hydroxylase (TH) decreased on the ipsilateral side to 6-OHDA microinjection. Scale bar represents 500 µm (low magnification) and 100 µm (high magnification). 3V: third ventricle. (**B**) Injection site into the A11 nucleus (gray circle: saline, black circle: 6-OHDA). Digit represents a distance from bregma (mm). (**C**) The mean number of TH immunoreactive (TH-IR) cells/section in the A11 nucleus (mean ± SEM). In ION-CCI rats with 6-OHDA microinjections, a significant decrease of TH-IR cells in the A11 nucleus was found in the ipsilateral side to 6-OHDA microinjection as compared with contralateral side. ION-CCI rats with 6-OHDA microinjections showed a significant decrease in TH-IR cells in the A11 nucleus on the ipsilateral side to 6-OHDA microinjection as compared with the ipsilateral side in ION-CCI rats with saline microinjections. *** *p* < 0.001. (**D**) The change in head-withdrawal threshold after microinjection of 6-OHDA. Data are shown as box and whisker plots. The bottom and top of the box indicate the first and third quartiles, respectively. The end of the whiskers represents the minimum and maximum values of all data. ION-CCI rats that received 6-OHDA microinjections into the A11 nucleus showed a significant decrease in head-withdrawal thresholds as compared with thresholds 7 days after ION-CCI (before 6-OHDA microinjections). ION-CCI rats that received 6-OHDA microinjections into the A11 nucleus showed a significant decrease in head-withdrawal thresholds as compared with ION-CCI rats that received saline microinjections 14 days after 6-OHDA microinjections (21 days after ION-CCI). *** *p* < 0.001 compared with head-withdrawal threshold before 6-OHDA microinjections. ^##^
*p* < 0.01 compared with head-withdrawal threshold in ION-CCI rats with saline microinjections. (**E**) Photomicrograph of pERK-IR cells in the Vc of ION-CCI rats that received microinjections of saline or 6-OHDA into the A11 nucleus. Scale bar represents 100 µm. (**F**) The mean number of pERK-IR cells/section in the Vc (mean ± SEM). The mean number of pERK-IR cells/section in the Vc was larger in ION-CCI rats that received microinjections of 6-OHDA as compared with ION-CCI rats that received microinjections of saline. * *p* < 0.05.

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
