# Peer review of "Dopaminergic Modulation of Orofacial Mechanical Hypersensitivity Induced by Infraorbital Nerve Injury"

_ijms, 2020, doi:10.3390/ijms21061945_

Round 1
Reviewer 1 Report
The paper by Hiroharu Maegawa et al. entitled “Dopaminergic modulation of orofacial mechanical 2 hypersensitivity induced by infraorbital nerve injury” reports A11 dopaminergic modulation of nociception in a rat model of orofacial mechanical hypersensitivity after infraorbital nerve injury. The overall findings are of interest for the field, the methods used are adequate and the obtained results are solid and coherent. However this manuscript does not represent a new advancement in the conceptual understanding of descending dopaminergic control of pain sensitivity. In addition analysis of results in cases of repeated measures appears to be incorrect.
Number of issues need to be addressed before accepting this publication (for details see comments below). The major problem with this manuscript is the lack of identification of a next significant question (considering dopaminergic control of pain sensitivity) which authors aim to resolve with their study. This is best reflected by the sentence in the Introduction: “However, the function of the A11 nucleus is not fully revealed” (Page 2 line 47). How the authors approach would provide better understanding is unclear. In fact, authors use very similar study design as in cited publications (use of D2 and GABAA agonist intra A11 microinfusions, 6-OHDA A11 lesions, measures of neuronal activity markers after painful stimulus - ie pERK - in the Vc). The one possible difference is the use of ION-CCI model of neuropathic pain (however Liu et al., 2019. PMID: 30325872) as in comparison to spinal nerve ligation reported previously. Thus, this manuscript provides further evidence of the modulatory role of A11 dopaminergic descending pathway in the regulation of pain sensitivity – which supports/corroborates previously published reports. This might be still considered as a significant contribution to the field – given that authors fix the issues detailed below.
Major comments:
- Data from Fig 1, Fig 2A, Fig 3B and Fig 4C should be analyzed with two-way repeated measures ANOVA with treatment and time factors as well as treatment × time interaction.
- For each statistical analysis please provide details such as F and df for each significant factor or interaction (or t and df in case of Student t test).
- The authors choose to use only one dose of each drug (quinpirole, muscimol) to study D2 or GABAA receptor role in the A11 in nociception while providing insufficient justification for the doses used. In addition, lack of additional doses prevents from assessing relationship between dose and effect thus reducing potential interpretations.
- Data reporting is confusing. While box graph could be very informative and its use could be applauded, for some reason authors do not show the data that they describe they will (ie page 13 lines 445-447) or show median (horizontal line in the box) with no description in the text. For example on Figure 1 the median is shown on 3 out of 10 boxes, whereas whiskers could be found only in 1 box and only signifying the data point with the lowest threshold (minimum value).
- All placements of the microinfusion cannula should be reported for each rat in each group. In addition number of misplacements should be reported.
- D2 as well as GABAA agonists decrease locomotor activity and cause sedation – which than could impact head withdrawal thresholds. Thus, intra-A11 administration of quinpirole or muscimol could result in increased withdrawal thresholds not due to modulation of nociception but locomotion and activity. Ideally, the same ION-CCI subjects could be used to study locomotor responses (e.g. vertical and horizontal activity) after intra-A11 administration. In addition, measures of nociception could be performed after application of mechanical stimuli to the maxillary whisker pad contralateral to the ION-CCI. These controls would rule out the potential sedation artifact. At minimum, such considerations should be discussed as a potential results interpretation confound.
- Lines 260-262 on page 10 are confusing and ambiguous. It could be interpreted that authors suggests that DA released in the A11 from the local DA neurons axon terminals (in the A11) diffuse to Vc to modulate Vc neurons activity via D2 receptors. Please clarify.
- The discussion lack a clear summary and authors proposition of how descending A11 DA pathway modulate pain, including neuropathic pain.
Minor comments:
- Please discuss the time-course of pERK induction in the Vc after mechanical stimuli in the ION-CCI model.
- Please report number of sections from which pERK induction was calculated.
- Please use abbreviation (e.g. CGRP) accordingly to journal prescriptions.
- Please include information about doses of drugs used on Figures.
- Line 306-307, page 11: please indicate what other (beside A11) descending dopaminergic pathway was shown by authors to modulate hypersensitivity in ION-CCI model. Alternatively, please rephrase.
Author Response
Major comments:
- Data from Fig 1, Fig 2A, Fig 3B and Fig 4C should be analyzed with two-way repeated measures ANOVA with treatment and time factors as well as treatment × time interaction.
Response
We performed two-way repeated measures ANOVA in the analyses of data from Fig 1, Fig 2A, Fig 3C (3B in previous manuscript) and Fig 4D (4C in previous manuscript). (page16 line 518, page 2 Figure 1, page 4 Figure 2A, page 7 Figure 3C, page 10 Figure 4D in manuscript without Track Changes; page 20 line 532, page 3 Figure 2, page 6 Figure 2A, page10 Figure 3C, page 14 Figure 4D in manuscript with Track Changes).
- For each statistical analysis please provide details such as F and df for each significant factor or interaction (or t and df in case of Student t test).
Response
We described details of each statistical analysis such as F and df. (page 2 line64-74, page 3 line 97-122, page 5 line 157- page 6 line 179, page 8 line 226- page 9 line 255 in manuscript without Track Changes; page 2 line 64-75, page 4 line 98-123, page 7 line 158- page 8 line 187, page 11 line 236- page 12 line 264 in manuscript with Track Changes).
- The authors choose to use only one dose of each drug (quinpirole, muscimol) to study D2 or GABAA receptor role in the A11 in nociception while providing insufficient justification for the doses used. In addition, lack of additional doses prevents from assessing relationship between dose and effect thus reducing potential interpretations.
Response
We agree reviewer1’s comment. In our study, we provided one dose not several doses for quinpirole or muscimol administration although doses of drugs were determined based on previous reports [15]. If several doses were selected for drugs administration, we might get more useful information.
- Data reporting is confusing. While box graph could be very informative and its use could be applauded, for some reason authors do not show the data that they describe they will (ie page 13 lines 445-447) or show median (horizontal line in the box) with no description in the text. For example on Figure 1 the median is shown on 3 out of 10 boxes, whereas whiskers could be found only in 1 box and only signifying the data point with the lowest threshold (minimum value).
Response
If no median is shown in the box, the median is at the top or bottom of the box. If whisker is not displayed in the box, the maximum and minimum values match the first and third quartiles.
- All placements of the microinfusion cannula should be reported for each rat in each group. In addition number of misplacements should be reported.
Response
We showed injection sites (Figure 3B, 4B) and described the number of misplacements. (page 5 line 154-157, page 8 line 190- 192, line 221-223, page 11 line 263-264, page7 Figure 3B, page 10 Figure 4B in manuscript without Track Changes; page 7 line 155-158, page 11 line 198-200, line 230-233, page 15 line 272-273, page 10 Figure 3B, page 14 Figure 4B).
- D2 as well as GABAA agonists decrease locomotor activity and cause sedation – which than could impact head withdrawal thresholds. Thus, intra-A11 administration of quinpirole or muscimol could result in increased withdrawal thresholds not due to modulation of nociception but locomotion and activity. Ideally, the same ION-CCI subjects could be used to study locomotor responses (e.g. vertical and horizontal activity) after intra-A11 administration. In addition, measures of nociception could be performed after application of mechanical stimuli to the maxillary whisker pad contralateral to the ION-CCI. These controls would rule out the potential sedation artifact. At minimum, such considerations should be discussed as a potential results interpretation confound.
Response
We agree reviewer1’s comment. We added following sentences to Discussion. Dopamine D2 agonist and GABAA receptor agonist were reported to affect motor function [33-36]. Therefore, the change of head-withdrawal threshold measured in our study might be result of not only attenuation of mechanical hypersensitivity but also change of motor function by quinpirole and muscimol. This point could be limitation of the present study. (page 12 line 351-355 in manuscript without Track Changes, page 16 line 362-366 in manuscript with Track Changes).
- Lines 260-262 on page 10 are confusing and ambiguous. It could be interpreted that authors suggests that DA released in the A11 from the local DA neurons axon terminals (in the A11) diffuse to Vc to modulate Vc neurons activity via D2 receptors. Please clarify.
Response
We revised corresponding section as follows; Taken together, these findings suggest that dopamine released from dopaminergic neurons in the A11 nucleus modulate the activity of the Vc neurons through dopamine D2 receptors expressed in the Vc. (page 12 line 320-322 in manuscript without Track Changes, page 16 line 329-331 in manuscript with Trach Changes).
- The discussion lack a clear summary and authors proposition of how descending A11 DA pathway modulate pain, including neuropathic pain.
Response
We revised corresponding section as follows; In summary, we showed that dopamine D2 agonist attenuated mechanical hypersensitivity induced by ION-CCI. And we suggested that the A11 nucleus is involved in the modulation of mechanical hypersensitivity induced by infraorbital nerve ligation in rats through a modulation of activities of the Vc neurons. (page 13 line 377-381 in manuscript without Track Changes, page 17 line 388-391 in manuscript with Trach Changes).
Minor comments:
- Please discuss the time-course of pERK induction in the Vc after mechanical stimuli in the ION-CCI model.
Response
We added following sections to Discussion. Following stimulation, pERK expression reaches to a peak within 3 min [19]. Previously, ION-CCI rats were perfused 5 min after stimulation, and pERK expression in the Vc was investigated [20]. Therefore, we perfused rats 5 min after mechanical stimulation to the maxillary whisker pad skin according to the previous report [20]. (page 11 line 290-293 in manuscript without Track Changes, page 15 line 299-302 in manuscript with Track Changes).
- Please report number of sections from which pERK induction was calculated.
Response
We added the number of sections selected for calculation to Material and Methods as follows. The five consecutive sections that contained the greatest number of pERK-IR cells in the Vc were selected in each rat (35 sections in each group) and the mean number of pERK-IR cells/section was calculated. (page 15 line 494-496 in manuscript without Trach Changes, page 19 line 508-510 in manuscript with Track Changes).
- Please use abbreviation (e.g. CGRP) accordingly to journal prescriptions.
Response
We corrected the use abbreviation accordingly to journal prescriptions. (page 1 line 14-15, page 12 line 345 in manuscript without Track Changes; page 1 line 14-15, page 16 line 356 in manuscript with Trach Changes)
- Please include information about doses of drugs used on Figures.
Response
We added information about doses of drugs. (page 7 line 188, page 11 line 259, page 15 line 464 in manuscript without Track Changes; page 11 line 196, page 15 line 268, page 19 line 478 in manuscript with Track Changes).
- Line 306-307, page 11: please indicate what other (beside A11) descending dopaminergic pathway was shown by authors to modulate hypersensitivity in ION-CCI model. Alternatively, please rephrase.
Response
Striatal administration of dopamine D2 receptor agonist was reported to attenuate withdrawal responses to mechanical stimuli in rats with peroneal and tibial nerve ligations [28]. Intrathecal administration of a dopamine D2 receptor agonist was reported to inhibit allodynic responses to mechanical stimuli in rats with sciatic nerve ligation [25]. When quinpirole was administered intraperitoneally, these regions might involve attenuation of mechanical hypersensitivity. However, pathways to attenuate mechanical hypersensitivity from these regions were not indicated from results of the present study. Therefore, we revised the corresponding sentence as follows; In summary, we show that the A11 nucleus is involved in the modulation of mechanical hypersensitivity induced by infraorbital nerve ligation in rats through a modulation of activities of the Vc neurons. (page 13 line 377-380 in manuscript without Trach Changes, page 17 line 388-391 in manuscript with Trach Changes).
Reviewer 2 Report
The manuscript titled: ‘Dopaminergic modulation of orofacial mechanical hypersensitivity induced by infraorbital nerve injury’ by Hiroharu Maegawa et al., provide some interesting results about the dopaminergic control system related to antinociception. I have some remarks to be explained.
Please, provide F values for the results.
Please, provide more details about type of pathology of chronic pain models using ION-CCI.
Discussion
The authors might consider referencing a study examining quinpirol in neuropathic models of pain in the rat Mercado-Reyes et al., 2019, assuming synergistic antinociceptive effect between opioid and dopaminergic system.
Author Response
Reviewer 2
Comments and Suggestions for Authors
The manuscript titled: ‘Dopaminergic modulation of orofacial mechanical hypersensitivity induced by infraorbital nerve injury’ by Hiroharu Maegawa et al., provide some interesting results about the dopaminergic control system related to antinociception. I have some remarks to be explained.
Please, provide F values for the results.
Response
We described F values in results of statistical analyses. (page 2 line 64-74, page 3 line 97-110, page 5 line 157-179, page 8 line 226-247 in manuscript without Track Changes; page 2 line 64-75, page 4 line 98-111, page 7 line 158-187, page 11 line 236-256 in manuscript with Track Changes)
Please, provide more details about type of pathology of chronic pain models using ION-CCI.
Response
We added following sections to Discussion. Previous report indicated that mechanical hypersensitivity induced by ION-CCI lasted 42 days after nerve ligation [41]. Heat hypersensitivity was also reported [27, 42], and it lasted 12 days after ION-CCI [42]. (page 13 line 370-372 in manuscript without Track Changes, page 17 line 381-383 in manuscript with Track Changes).
Discussion
The authors might consider referencing a study examining quinpirol in neuropathic models of pain in the rat Mercado-Reyes et al., 2019, assuming synergistic antinociceptive effect between opioid and dopaminergic system.
Response
We added following sections to Discussion. Previous report indicated that intrathecal coadministration of quinpirole and µ-opioid receptor agonist attenuated mechanical and heat hypersensitivity induced by sciatic nerve ligation [43]. Increases of dopamine D2 receptors and their colocalization with µ-opioid receptors were reported in the dorsal horn during pathological process of pain [44, 45]. Similar reaction may occur in the Vc of ION-CCI rats. (page 13 line 372-376 in manuscript without Track Changes, page 17 line 383-387 in manuscript with Track Changes).
Round 2
Reviewer 1 Report
The revised paper by Hiroharu Maegawa et al. entitled “Dopaminergic modulation of orofacial mechanical hypersensitivity induced by infraorbital nerve injury” addressed most of previously raised concerns however several issues needs to be resolved (see comments below), in particular in regard of correct use of statistical analysis.
- The interpretation of statistical analysis is incorrect. When using two-way ANOVA and two-way repeated-measures ANOVA, if the interaction between factors is significant (e.g. Fig. 1), it is incorrect to interpret and there is no need to report stats for main effects (even when these effect are significant). Report only the stats for the interaction (F, df, p). In order to interpret significant interaction, the post hoc tests should be conducted and p value for specific post hoc tests should be reported. In other cases, please report the stats for all factors and interaction. In addition, if the main effects are significant (but the interaction is not), run the post hoc test for the comparison between the groups. Finally, in the methods section, please provide which post hoc comparison has been used (Bonferoni?).
- Please provide description in the methods section of the median measure in the box graphs as well as explanation why in some cases there is no median or whiskers.
- Please increase quality of graphs showing histological verifications in Figures 3 and 4. I suggest adapting graphs from the Paxinos atlas of rat brain.
Author Response
Response to reviewer1’s comments
- The interpretation of statistical analysis is incorrect. When using two-way ANOVA and two-way repeated-measures ANOVA, if the interaction between factors is significant (e.g. Fig. 1), it is incorrect to interpret and there is no need to report stats for main effects (even when these effect are significant). Report only the stats for the interaction (F, df, p). In order to interpret significant interaction, the post hoc tests should be conducted and p value for specific post hoc tests should be reported. In other cases, please report the stats for all factors and interaction. In addition, if the main effects are significant (but the interaction is not), run the post hoc test for the comparison between the groups. Finally, in the methods section, please provide which post hoc comparison has been used (Bonferoni?).
Response
We appreciate you for reading our manuscript and giving useful comments again. We revised interpretation of statistical analyses according to your comment. (page 2 line 64-73, page 3 line 93-103, page 6 line 146- page 7 line 169, page 10 line 218- page 11 line 231, page 19 line 509 in manuscript with Track Changes; page 2 line 64-72, page 3 line 92-102, page 5 line 145-162, page 8 line 208- page 9 line 222, page 16 line 493 in manuscript without Trach Changes).
- Please provide description in the methods section of the median measure in the box graphs as well as explanation why in some cases there is no median or whiskers.
Response
We added following section into Material and Methods. If no median is shown in the box, the median is at the top or bottom of the box. If whisker is not displayed in the box, the maximum and minimum values match the first and third quartiles. (page 19 line 505-507 in manuscript with Trach Changes, page 16 line 489-491 in manuscript without Track Changes).
- Please increase quality of graphs showing histological verifications in Figures 3 and 4. I suggest adapting graphs from the Paxinos atlas of rat brain.
Response
We revised Figure 3B and 4B by adapting graphs from the Paxinos atlas of rat brain. (page 9 and 12 in manuscript with Track Changes, page 7 and 9 in manuscript without Track Changes).